# Pancreatic Adenocarcinoma: Imaging Modalities and the Role of Artificial Intelligence in Analyzing CT and MRI Images

**DOI:** 10.3390/diagnostics14040438

**Published:** 2024-02-16

**Authors:** Cristian Anghel, Mugur Cristian Grasu, Denisa Andreea Anghel, Gina-Ionela Rusu-Munteanu, Radu Lucian Dumitru, Ioana Gabriela Lupescu

**Affiliations:** 1Faculty of Medicine, Department of Medical Imaging and Interventional Radiology, Carol Davila University of Medicine and Pharmacy Bucharest, 020021 Bucharest, Romania; anghel.cristians@yahoo.com (C.A.); radu.dumitru@umfcd.ro (R.L.D.); ioana.lupescu@umfcd.ro (I.G.L.); 2Department of Radiology and Medical Imaging, Fundeni Clinical Institute, 022328 Bucharest, Romania; denisa_calinescu@yahoo.com (D.A.A.); gina.m.rusu@icfundeni.ro (G.-I.R.-M.)

**Keywords:** pancreatic adenocarcinoma, artificial intelligence, pancreas imaging

## Abstract

Pancreatic ductal adenocarcinoma (PDAC) stands out as the predominant malignant neoplasm affecting the pancreas, characterized by a poor prognosis, in most cases patients being diagnosed in a nonresectable stage. Image-based artificial intelligence (AI) models implemented in tumor detection, segmentation, and classification could improve diagnosis with better treatment options and increased survival. This review included papers published in the last five years and describes the current trends in AI algorithms used in PDAC. We analyzed the applications of AI in the detection of PDAC, segmentation of the lesion, and classification algorithms used in differential diagnosis, prognosis, and histopathological and genomic prediction. The results show a lack of multi-institutional collaboration and stresses the need for bigger datasets in order for AI models to be implemented in a clinically relevant manner.

## 1. Introduction

Pancreatic ductal adenocarcinoma (PDAC) holds the distinction of being the most prevalent and aggressive form of pancreatic malignancy, with a five-year relative survival rate of 12% [1]. According to the American Cancer Society, about 66,440 people will be diagnosed with pancreatic cancer and about 51,750 people will die of pancreatic cancer in the United States in 2024 [2]. It is currently the fourth leading cause of all cancer deaths in both genders, with data suggesting that it will become the second leading by 2030 [3,4,5,6,7]. The most important risk factors for developing PDAC are tobacco use, obesity, personal history of diabetes or chronic pancreatitis, family history of pancreatic cancer or pancreatitis, and certain hereditary conditions (such as Peutz-Jeghers syndrome, von Hippel-Lindau syndrome, hereditary breast and ovarian cancer syndrome, and multiple endocrine neoplasia type 1 syndrome) [8,9]. PDAC is in most cases diagnosed in advanced stages, with only a minority of patients having a resectable disease. Early-stage diagnosis remains a challenge because most of the patients have nonspecific symptoms or are asymptomatic until the tumor reaches an advanced stage.

Pancreatic tumors can be detected through ultrasound and magnetic resonance (MRI) images, but the principal modality for assessing pancreatic adenocarcinoma is contrast-enhanced computer topography (CECT) due to its ability to correctly define the extent of tumor infiltration (especially the blood vessels) and the existence of metastases [10]. Based on the imaging features at diagnosis, patients with PDAC are generally classified into resectable, borderline resectable, unresectable, and metastatic disease.

Artificial intelligence (AI) has gained a great interest in oncology because it can integrate high amounts of data to produce personalized recommendations based on each patient’s clinical and paraclinical characteristics (serum markers, gene expression, imaging features) [11]. With increasing medical images available from CT and MRI scans, there is a trend toward using different imaging features to train artificial intelligence algorithms in the classification process (for example, in differentiating a benign condition from a malignant tumor), which could help the medical team to decide if the patient needs to undergo surgery or not. This tool could help the radiologist to better interpret the images, especially in difficult cases. However, one of the challenges in developing an AI model is the need for a big dataset, so that the model can be trained on a high number of cases.

Radiomics aims to extract quantitative information from diagnostic images (CT, MRI, PET, SPECT, or ultrasound), including complex patterns that the human eye cannot recognize [12,13]. Analyzing the tissue characteristics based on shape, size, volume, and texture can offer information on tumor histopathology and microenvironment [14]. Radiogenomics represents the correlation between the genomic pattern and the radiomic data. The amount of information extracted from a voxel level can surpass the routine radiology interpretation. Identifying the changes and characteristics that are present at a voxel level could open the way for multiple correlations such as with the histopathological grade and the presence of different genes and mutations. Oncologic patients undergo multiple imaging examinations, and in the near future, there will likely be available large data sets comprising images of different histological types, enabling the collection of radiomic features with the purpose of creating better treatment algorithms and increasing overall survival.

The detection of PDAC on CT or MRI images at small dimensions can be difficult, but with an AI algorithm trained on a large dataset of pancreatic tumors, we believe that the accuracy in detecting small tumors can improve. The key in the management of pancreatic tumors is represented by the detection of lesions at small dimensions (even <1 cm), making the tumor resectable.

The segmentation of the tumor by implementing a supervised AI model verified by a radiologist could help to better delineate the tumor margins in regard to the major vessels and could improve surgical planning. Tumor segmentation will be further needed in the classification process.

There are some free software tools available that could be used by researchers for studying radiomics features in PDAC (Table 1). We believe that by making AI tools available and sharing datasets with different pathologies, more and more researchers will start to study the role of AI in analyzing medical images. There are some free datasets available for download, most of them used in challenges, such as Medical Segmentation Decathlon [14,15], AbdomenCT-1K [16] and Pancreas-CT [17], but hopefully in the future, more of them will be available.

## 2. Imaging Modalities

CECT emerges as the preferred imaging modality for assessing PDAC, owing to its capacity to delineate vascular contacts with major blood vessels and to identify metastatic lesions, primarily within the liver, lung, and peritoneum [10]. The National Comprehensive Cancer Network (NCCN) guideline recommends the use of CECT as the preferred modality for evaluation at presentation and four weeks before surgery and following neoadjuvant treatment for the staging and assessment of resectability status [27]. The typical pancreatic CECT protocol is composed of dual-phase acquisition after intravenous contrast medium administration with a pancreatic phase around 35–40 s after contrast agent injection and a portal venous phase after a 65 to 70 s delay [28]. The advantage of the pancreatic phase consists of better contrast between the increased enhancement of the pancreatic normal parenchyma and the hypoattenuating adenocarcinoma. The portal venous phase allows the easier detection of the hepatic metastases and venous thrombus. One way to improve the enhancement of pancreatic parenchyma and adjacent blood vessels is to use a saline flush in addition to power injection [29]. Resectability criteria are shown in Table 2 [27,30]. An example of PDAC diagnosed at CECT is shown in Figure 1.

While magnetic resonance imaging (MRI) boasts superior soft tissue contrast resolution, it does not surpass computed tomography (CT) in the staging of PDAC [31]. According to some studies, MRI is useful in characterizing smaller and isoattenuating tumors on CT, as well as distinguishing focal fatty infiltration from a tumor [31,32]. The MRI protocol recommended by the NCCN for the evaluation of PDAC consists of the following: T2-weighted imaging (WI) single-shot fast spin-echo (SSFSE) in coronal and/or axial planes; T1WI in-phase and out-phase gradient echo (GRE) in the axial plane; T2WI fat-suppressed fast spin-echo (FSE) in the axial plane; diffusion weighted imaging (DWI) in the axial plane; pre- and dynamic post-intravenous contrast administration 3D T1WI fat-suppressed GRE in the pancreatic, portal venous, and equilibrium phases, in the axial plane; and T2WI cholangiopancreatography (MRCP) in the coronal plane [27,33]. An example of PDAC diagnosed via MRI is shown in Figure 2. A study by Fang-Ming Chen et al. that compared CT and MRI in the presurgical evaluation of pancreatic cancer in 38 patients showed that the evaluation of vessel involvement, nodal status, and resectability had no statistically significant differences between CT and MRI [34].

Abdominal ultrasound is a non-invasive diagnostic technique that may be used to diagnose pancreatic lesions. However, it is an operator-dependent modality that, in most cases, does not allow an optimal view and characterization of the entire pancreas because of the gas interposition artefacts. The sensitivity of US (ultrasonography) in detecting pancreatic adenocarcinoma varies from around 50% to 90% [35,36].

## 3. Methods

Our objectives are to review the current imaging protocols for PDAC and to provide an overview of the current AI applications researched in this pathology. There is an increased interest in AI applications in PDAC that is reflected in the number of papers published in the last five years. We performed an advanced search on PubMed for articles published in English containing the keywords “pancreatic adenocarcinoma” and “artificial intelligence”. The graph below shows the number of papers published per year with these specific filters (Figure 3). A total of 310 papers were identified and further screened. We excluded the papers that focused on other imaging modalities than CT and MRI (ultrasound, PET-CT, echo-endoscopy) and the ones that described AI models used in other instances (such as histopathology or laboratory data). A total of 59 papers were selected, from which 20 were reviews and the other 39 were original research studies (Table A1). We decided to divide the articles into 3 categories: detection, segmentation, and classification. We discussed the findings in the upcoming section.

## 4. AI Applications in PDAC

### 4.1. Detection

PDAC carries a poor prognosis because of the delayed diagnosis and the lack of screening modalities and protocols in the general population. Studies indicate that the detection of PDAC in high-risk individuals leads to a median overall survival of 9.8 years (compared to 1.5 years for those diagnosed outside of surveillance) [37]. This can be correlated to the fact that early detected tumors are smaller in size and do not infiltrate the surrounding vessels. Identifying small and iso-attenuating tumors can be sometimes difficult, especially in asymptomatic patients, and can lead to delayed recognition [38,39]. There is a great interest in developing an AI model that could identify PDAC on prediagnostic CTs before the clinical diagnosis. AI algorithms show promise in detecting subtle changes in pancreas and small pancreatic tumors, that can be missed by human observers.

After reviewing the latest articles that focused on AI algorithms for the detection of PDAC we believe that identifying tumors at small dimensions is crucial for having a resectable tumor, and the main focuses of AI applications should be the detection of small pancreatic tumors (preferably <2 cm). Because most of the tumors are large at diagnosis, the AI models are trained on data sets that contain a small proportion of small tumors, making it hard for the model to achieve good results in detecting small and isoattenuating PDAC. Large datasets that contain an increased number of small tumors should be made publicly available in order for researchers to be able to train their models on and have better results in detecting small (even <1 cm) tumors in high-risk patients. Selected papers that focused on detection can be seen in Table 3. 

Cao et al. present a novel deep learning framework named Pancreatic Cancer Detection with Artificial Intelligence (PANDA) aimed at detecting and categorizing pancreatic lesions utilizing non-contrast CT images [40]. This model underwent training using a dataset comprising non-contrast CT scans from 3208 patients sourced from a single center. In a multicenter validation encompassing 6239 patients from 10 distinct centers, the model demonstrated exceptional performance with an area under the receiver operating curve (AUC) ranging between 0.986 and 0.996 for lesion detection. Their model could represent a screening method in the high-risk population, given the lower dose compared to CECT and the absence of risk given by the administration of intravenous contrast.

Korfiatis et al. conducted a case–control study to develop an automated 3-dimensional (3D) Convolutional Neural Network (CNN) for detecting PDAC on diagnostic computer tomography scans [41]. One of their primary aims was to assess the model’s capability to identify visually imperceptible preinvasive cancer on prediagnostic CT scans. The 3D-CNN was trained using a dataset consisting of 696 portal phase diagnostic CT scans featuring PDAC cases and 1080 control images displaying non-neoplastic pancreatic conditions. Despite being exclusively trained on CT images depicting larger tumors, the model successfully detected occult PDAC on prediagnostic CT scans (incidentally acquired 3–36 months before the clinical diagnosis of PDAC) with AUC curve 0.91, sensitivity 0.75, and specificity 0.90 at a median of 475 days before clinical diagnosis. Their model shows great promise in future screening of high-risk individuals and could alert readers for small lesions and secondary signs of PDAC (atrophy and ductal dilatation), but we believe that further training on CTs images with smaller tumors will increase the sensitivity and the specificity. 

Chu et al. engineered a deep-learning algorithm designed to differentiate CT cases from patients afflicted with PDAC versus those from control subjects [42]. Initial findings from an analysis comprising 156 PDAC cases and 300 normal cases revealed a sensitivity of 94.1% and a specificity of 98.5% for the detection of PDAC. The deep network performed well on larger tumors and often missed small tumors, but this problem could be overcome by adding more small tumors in the training data set and modifying the model structure to focus on finding smaller lesions [42]. Their model shows good results, but needs to be validated on larger data sets, including images from other institutions.

Another study determined the utility of radiomics features in differentiating CT cases of PDAC from normal pancreas [43]. In this study, 190 patients with PDAC and 190 healthy potential renal donors were included, with 40 radiomic features selected for analysis. The dataset was partitioned into 255 training cases, comprising 125 normal control cases and 130 PDAC cases, and 125 validation cases, including 65 normal control cases and 60 PDAC cases. The random forest binary classification achieved an accuracy of 99.2% and an AUC of 99.9%. However, it is noteworthy that the mean tumor size within this dataset was 4.1 cm, a dimension typically within the range of detection capability for the average radiologist. The results need to be validated on datasets containing much smaller tumors (even as small as 1–2 cm) in order to have a clinical impact and assist the radiologist in detecting tumors while still at a resectable stage.

Chu et al. have also published a paper comparing the diagnostic performance of commercially available vs. in-house radiomics software in the classification of CT images from patients with PDAC vs. healthy controls [44]. When employing 40 features in the random forest classification, the in-house software demonstrated superior sensitivity at 100% and accuracy at 99.2%, surpassing the performance of the commercially available research prototype, which exhibited a sensitivity of 95% and accuracy of 96.8%. However, when the number of features was reduced to five, the diagnostic performance of the in-house software decreases to sensitivity (95%) and accuracy (93.6%), while the diagnostic performance of the commercially available prototype was unchanged. The study population was selected based on a previously published study [43]. Future studies are needed to determine if other commercially available radiomics software will achieve similar results.

Ma et al. constructed a CNN model using a dataset of 3494 CT images from 222 patients confirmed with PDAC and 3751 images from 190 patients with normal pancreas with the aim to distinguish pancreatic cancer from benign tissue [45]. The plain phase had the same sensitivity (91.58%) as the arterial and venous phases, with an accuracy of 95.47% and specificity of 98.27%. Given the lower radiation dose and easier access to non-contrast CT, their model is suitable for PDAC screening, especially in patients with contraindications for the intravenous administration of contrast media.

Alves et al. developed a deep learning model used for PDAC detection, focusing on small lesions [46]. The study included CECT images from 119 pathology proven PDAC patients and 123 patients without PDAC that were used to train a nnUnet for automatic lesions detection and segmentation. The proposed model achieved good results with a maximum AUC of 0.914 in the whole external test and 0.876 in the subgroup of tumors < 2 cm. However, their study included only tumors that were located in the pancreatic head and further studies are needed on data sets including lesions located in the body and tail of the pancreas.

Mukherjee et al. developed a radiomics-based machine-learning (ML) model to detect PDAC at the prediagnostic stage [47]. They included prediagnostic CTs (median interval between CT and PDAC diagnosis: 398 days) of 155 patients and an age-matched cohort of 265 patients with normal pancreas, with 34 radiomics features selected. An internal set of 176 patients and 80 publicly available control cases were used for validation. A machine learning classifier, support vector machine (SVM), had a sensitivity of 95.5% and AUC of 0.98. Their study shows good results in detecting PDAC at a prediagnostic stage, which could greatly benefit the patients by identifying the tumors in a resectable stage.

Wang et al. addressed the issues of consistency and reproducibility of feature extraction (radiomics) and subsequent modeling by exploring the resilience of radiomics features against various sources of uncertainty including bin width resampling, image transformation, image noise, and segmentation growth/shrinkage. [48]. They collected venous-phase CECT images from 181 control subjects and 85 PDAC case subjects and extracted 924 radiomics features using PyRadiomics. The training set was formed from 189 cases (60 cancer and 129 control) and the remaining 77 cases (25 cancer and 52 control) formed the testing set. They trained a Random Forest model using eight nonredundant radiomics features to distinguish the cancer patients from healthy individuals, which achieved a mean AUC of 0.99 ± 0.01 on the training dataset by cross-validation and an AUC of 0.910 on the independent data set. However, the sample size was limited compared to other published studies [43,49]. Their model needs validation on bigger data sets.

Viviers et al. proposed a model based on a U-Net-Like Deep CNN that incorporates the external secondary features such as the pancreatic duct, common bile duct, and the pancreas, in conjunction with a processed CECT scan [50]. The study included a dataset of 99 cases of PDAC located in the pancreatic head and 97 control cases without a pancreatic tumor. The model achieved a performance score for classification and localization of 99% sensitivity and 99% specificity. However, the CNN model cannot discover the causal relationship between these secondary features and the presence of the tumor, with future work needed to enable CNN to fully integrate all the information. The study focuses on secondary features of the PDAC, which is of great interest especially in isoattenuating and small tumors. Their model also included only patients with PDAC located in the pancreatic head, with further research needed to see if the model obtains the same results for tumors located in the pancreatic body or tail.

**Table 3 diagnostics-14-00438-t003:** AI detection models.

Author	Year	Modality	Approach	Sensitivity
Cao et al. [40]	2023	Non-contrast CT	Deep learning	92.9%
Korifatis et al. [41]	2023	CECT	3D-CNN	75%
Alves et al. [46]	2022	CECT	CNN	N.A.
Wang et al. [48]	2022	CECT	Radiomics	84%
Mukherjee et al. [47]	2022	CECT	Radiomics-ML	95.5%
Viviers et al. [50]	2022	CECT	U-Net-Like Deep CNN	99%
Ma et al. [45]	2020	Non-contrast CT	CNN	91.58%
Chu et al. [42]	2019	CECT	Deep learning	94.1%
Chu et al. [43]	2019	CECT	Radiomics	100%

### 4.2. Segmentation

Most of the AI radiology models are supervised and need large amounts of images for training [51]. Image preparation including lesion and organ segmentation is performed in most instances manually or in a semi-automated manner by the radiologist. This process needs a huge amount of work, and given the fact that for AI models to be validated large datasets are needed for training and testing, a great interest in the automatic segmentation of the pancreas and pancreatic lesions is seen in the last years. An example of semi-automated tumor segmentation using 3D Slicer software ver. 5.6.1 [22] is shown in Figure 4.

The automated segmentation of PDAC in CT scans represents a challenge given (1) the variability in size, shape, and location of the tumor in the pancreas, (2) the small size of the pancreas and the tumor by comparison with the entire abdomen and (3) the poor contrast of the pancreas with the surrounding structures [52,53,54]. AI segmentation algorithms include multiorgan atlas-based, landmark-based, shape model-based, and neural network-based [55]. The Dice similarity coefficient (DSC) is used to evaluate the overlap between a predicted segmentation mask and the ground truth segmentation (performed manually by the radiologist), with a score that ranges from 0 (no spatial overlap between two sets of binary segmentation results) to 1 (complete overlap) [55]. Selected papers that focused on segmentation can be seen in Table 4.

Ni et al. investigated the reliability of radiomic features extracted from CECT by AX-Unet, a pancreas segmentation model that integrates the strengths of deepLabV series [56], Unet, and Xception networks [57], to analyze the recurrence of PDAC after radical surgery [58]. The model was trained on a set of 205 PDAC patients and evaluated on an independent testing set of 64 patients with clear prognoses. The Dice score for their dataset reached 85.9%, showing great effectiveness for PDAC image segmentation. The study shows good results, with great potential in accurately identifying and segmenting the tumor and maximizing the tumor surgical removal.

Turečková et al. proposed the extension of CNN by including deep supervision and attention gates [59]. The model examined a dataset comprising 421 portal venous-phase CT images and corresponding segmentation masks delineating the pancreas and tumor, collected from the Memorial Sloan Kettering Cancer Center. Among these, 282 images were included in the training set, which was publicly released for the 2018 MICCAI Medical Decathlon Challenge [14]. During training, they conducted five-fold cross-validation, employing 26 images for validation and 105 images for training. The Dice scores achieved by their VNet-AG-DSV model were 81.22% for pancreas labels and 52.99% for tumor labels, respectively. In comparison, Isensee et al. achieved in the MICCAI Medical Decathlon Challenge a Dice score for the pancreas and the pancreas tumor segmentation of 79.30% and 52.12% using a nnU-Net model [60]. The findings show great potential in incorporating deep supervision and attention gates for pancreatic segmentation; however, we believe that further validation of bigger data sets should be performed.

The algorithms studied for segmenting PDAC are optimized for only one type of input (one phase of CT acquisition); therefore; the models cannot be implemented in a multi-phase instance. To address this problem, Zhou et al. proposed a multi-phase segmentation model, Hyper-Pairing Network (HPN) for better performance for detecting pancreatic tumors [53]. They constructed a dual-path network for handling multi-phase data and applied skip connections across different paths of the network (referred to as hyper-connections) to enable information exchange between different phases. They also applied an additional pairing loss that encourages the commonality between the two sets of high-level semantic representations to reduce view divergence. The model was trained on arterial and venous phase CT images from 239 patients with pathologically proven PDAC. Compared to single-phase algorithms, multi-phase algorithms had a superior DSC (57.10% ± 24.76% for 3D-UNet-multi-phase-HPN and 63.94% ± 22.74% for 3D-ResDSN-multi-HPN). This study opens the pathway for further research on multi-phase segmentation, that is much needed in PDAC diagnosis because some tumors have a heterogeneous appearance and could benefit from both arterial and venous phase analysis.

Mahmoudi et al. used a 3D-CNN architecture localized in the pancreas region from the whole CT volume using 3D Local Binary Pattern (LBP) map of the original image [61]. The pancreatic tumor was subsequently segmented using 2D attenuation U-Net and Texture Attention U-Net (TAU-Net) that was introduced by the fusion of dense Scale-Invariant Feature Transform (SIFT) and LBP descriptors intro the attention U-Net. An ensemble model was used to summarize the advantages of both networks using a 3D-CNN. The Dice global score for the proposed hybrid model auto-localization was 60.6%. The model showed a good performance in segmentation on small sample size; however, further research is needed on larger data sets for this algorithm to be validated.

Most of the proposed algorithms for PDAC segmentations have been trained on large tumors, but there is little research focused on the early signs of PDAC (such as dilated pancreatic duct). Shen et al. addressed this problem and trained a fully convolutional networks (FCN) to generate an ROI covering the pancreas and used a 3D U-Net-like FCN for coarse pancreas segmentation [62]. The average Dice score and sensitivity was 49.9% and 51.9%, respectively. Their work shows the potential of using AI models for dilated pancreatic duct segmentation, which could also be beneficial in the detection of small tumors.

One of the biggest inconveniences that we found in the studies that focused on PDAC was the lack of an automated pancreatic or tumor segmentation model. Segmentation is time-consuming, needs an experienced radiologist, and is not feasible in a clinical setting. Developing automated segmentation AI software should make research less time-consuming and could increase the number of patients in the datasets, improving the results. However, the studies that we found focused on segmentation on CECT images, but given the increasing number of MRI machines available in the world, we believe that further studies need to focus on segmentation on different MRI sequences, making the models applicable both on CT and MRI images.

**Table 4 diagnostics-14-00438-t004:** AI segmentation models for PDAC.

Author	Year	Modality	AI Method	DICE Score
Ni et al. [58]	2023	CECT	AX-Unet	85.9%
Mahmoudi et al. [61]	2022	CECT	Hybrid 3D-CNN model	60.6%
Turečková et al. [59]	2020	CECT	VNet-AG-DSV	52.99%
Zhou et al. [53]	2019	CECT	3D-ResDSN-multi-HPN	63.94%
Isensee et al. [60]	2018	CECT	nnU-Net	52.12%

### 4.3. Classification

#### 4.3.1. Differential Diagnosis

One IA research area in PDAC represents the use of algorithms in differential diagnosis with other benign or maligning conditions. This could be of great use because in some instances the treatment is different and could improve patient management by avoiding unnecessary surgery. Selected papers that focused on differential diagnosis can be seen in Table 5.

Focal-type autoimmune pancreatitis (fAIP) represents a segmental involvement of the pancreatic parenchyma and can mimic PDAC, making the differential diagnosis very difficult. Most patients with fAIP will undergo surgical resection for a clear histopathological diagnosis. However, with the development of new AI models, there is hope that a clear discrimination between these entities will be achieved, eliminating the need for patients to undergo future surgical resection. Lu et al. developed a CT-based radiomics nomogram combining subjective CT findings and radiomic features for the preoperative differentiation of fAIP from PDAC [63]. The study included CT images from 96 patients (32 with AIP and 64 with PDAC) with 158 features extracted from each image. Seven features were selected by the LASSO algorithm and a multiparametric radiomics signature was established. They constructed a combined model of radiomics nomogram using multivariate logistic analysis of five characteristics (capsule-like rim, pancreatic atrophy, biliary wall thickening, vascular invasion, and Rad-Score). The nomogram model exhibited robust sensitivity and specificity, achieving AUC values of 0.87 and 0.83 in the training and test cohorts, respectively. The study had some limitations, one of them being the low number of fAIP cases acquired over a period of 11 years (2011 to 2021), which are not enough to validate the proposed radiomics model. However, further studies on larger data sets from multiple centers could improve the differential diagnosis between PDAC and fAIP.

Another study was conducted by Park et al. on differentiating autoimmune pancreatitis (AIP) from PDAC with radiomics features that included CECT images from 89 patients with AIP and 93 patients with PDAC [64]. Using the CT radiomics features in the test set, AIP could be distinguished from PDAC with a sensitivity of 89.7% (95% CI: 78.6–100%), a specificity of 100% (95% CI: 93–100%), an accuracy of 95.2% (95% CI: 89.8–100%), and an AUC of 0.975 (95% CI: 0.936–1.0). Limitations of the study were the small sample size that included both focal and diffuse forms of AIP and the period of time in which the images were acquired for the patients with AIP (2004 to 2018).

Anai et al., constructed a support vector machine (SVM) classifier using CT textured analysis for differentiating fAIP and PDAC and to assess the radiologist’s (4) diagnostic performance with or without SVM [65]. The study included CECT images from 20 patients with fAIP and 30 patients with PDAC from which 62 texture-based features were extracted. The SVM classifier had strong performance in differentiating fAIP and PDAC, with an AUC of 0.920. The SVM classifier increased the AUC for all four readers, but the reader with less experience benefitted the most from the SVM outputs. However, given the small number of patients included in the study, there is need for a larger study population to confirm the results.

Mass-forming chronic pancreatitis (MFCP) is another benign condition that can easily mimic PDAC lesions. A study addressed the problem of discriminating between PDAC and mass-forming chronic pancreatitis lesions (MFCP) [66]. This retrospective study included 119 patients from two independent institutions that had performed preoperative MRI. They extracted four feature sets from T1-weighted imaging (T1WI), T2-weighted imaging (T2WI), arterial phase, and portal phase of dynamic contrast-enhanced MRI, and radiomics models were constructed. The study shows good results with AUCs for the T1WI, T2WI, arterial and portal phases of 0.893, 0.911, 0.958, and 0.997 in the primary cohort and 0.882, 0.902, 0.920, and 0.962 in the validation cohort. We can see that the best result was obtained on the portal phase of dynamic contrast-enhanced MRI. Some limitations of the study are the small sample size and the fact that the analysis of the area of interest was performed in a two-dimensional manner rather than a three-dimensional analysis of the entire volume of the lesion. The study is of great interest because it was performed on MRI images, in contrast to most of the studies that are performed on CT images.

Ren et al. proposed a CT-based radiomics signature for the differential diagnosis between pancreatic adenosquamous carcinoma (PASC) and PDAC [67]. They used CECT images (both late arterial phase and portal venous phase) from 81 patients with PDAC and 31 patients with PASC to extract radiomics features. Seven radiomics features from late arterial phase images and three from portal venous phase images were selected and a radiomics signature was constructed. Using a 10-times leave group our-cross-validation method (LGOCV), the radiomics signature was proven to be robust and reliable with an average AUC of 0.82. However, the limitations of the study were the small number of PASC patients, different scanners used for image acquisition, and a selection bias due to retrospective nature of the study.

Shi et al. evaluated the performance of the histogram array and CNN based on diffusion-weighted imaging (DWI) with multiple b-values to distinguish PDAC from solid pseudopapillary neoplasm (SPN) and pancreatic neuroendocrine neoplasm (PNEN) [68]. They included 231 patients (132 PDACs, 45 PNENs, and 54 SPNs), which were further divided into a training group (n = 92), validation group (n = 46), and testing group (n = 93). The DWI images from 10 b values were converted into a histogram array that was entered into the CNN model. The AUCs of CNN for distinguishing PDACs from non-PDACS were 0.896, 0.846, and 0.839 in the training, validation, and testing groups. One of the limitations of the study was the low number of PNENs and SPNs used for training the CNN models, given the fact that these pancreatic tumors are rare compared to PDAC [69]. The study also did not include other types of pancreatic tumors such as metastases, mucinous tumors, or lymphoma. We posit that the methodology used in the study is of great interest and further research should also include in the differential diagnosis other entities while validating the results on larger data sets.

Zhang et al. evaluated the diagnostic efficacy of radiomics in conjunction with multiple machine learning algorithms for distinguishing between PDAC and pancreatic neuroendocrine tumors (pNET) [70]. This was achieved by constructing 45 discriminative models using five selection algorithms and nine classification algorithms. [70]. The study included CECT images from 238 patients (156 diagnosed with PDAC and 82 diagnosed with pNET). By using the combination of Gradient Boosting Decision Tree (GBDT) as the selection algorithm and Random Forest (RF) as the classification algorithm, the model reached the highest AUC: 0.971 in the training set and 0.930 in the validation set. The study reinforces the idea that multi-algorithm modeling should be considered when predicting subtypes of cancer with machine learning.

Current research that analyzed the differential diagnosis of PDAC from other types of pancreatic lesions has been conducted in a binary fashion (distinguishing PDAC from other types of pancreatic lesions, for example, neuroendocrine tumors); however, for a model to be clinically relevant it should be able to distinguish between a more complex number of pathologies (for example, autoimmune pancreatitis, chronic pancreatitis, cystic lesions, and other malignancies). We contend that in the future, research should focus on the detection of the correct classification among a variety of pancreatic diseases.

**Table 5 diagnostics-14-00438-t005:** AI models studied in differential diagnosis of PDAC.

Author	Year	Modality	Scope	Approach	AUC
Lu et al. [63]	2023	CECT	fAPI vs. PDAC	Radiomics	0.83
Shi et al. [68]	2023	MRI	PDAC vs. PNEN and SPN	CNN	0.839
Zhang et al. [70]	2022	CECT	PDAC vs. pNET	Radiomics	0.930
Anai et al. [65]	2022	CECT	fAPI vs. PDAC	Radiomics	0.920
Deng et al. [66]	2021	MRI	MFCP vs. PDAC	Radiomics	0.962
Ren et al. [67]	2020	CECT	PASC vs. PDAC	Radiomics	0.82
Park et al. [64]	2020	CECT	AIP vs. PDAC	Radiomics	0.975

#### 4.3.2. Histopathological Subtype and Genomic Features

The histopathological subtype is a crucial factor in PDAC prognosis, with poorly differentiated tumors showing higher aggression and shorter survival than well-differentiated PDAC [71,72]. Poor differentiation represents an independent prognostic factor that affects overall survival in patients with PDAC [73]. Knowing the histopathological grade before the surgical intervention might change the treatment options for patients with poorly differentiated tumors; however, the histopathological grade is often determined after surgery. It still remains a challenge to determine the histopathological grade of the tumor due to marked morphological tumor heterogeneity and the limited amount of tumor tissue samples in cases of tumor biopsy [74,75,76,77]. Developing a noninvasive technique to differentiate between the subtypes of PDAC before treatment could help to maximize patient survival and avoid the risks associated with surgical resection. Selected papers that focused on histopathological subtype and genomic features can be seen in Table 6.

Qiu et al. constructed a score using a support vector machine (SVM) to predict the histopathological grade (low/high-grade) of PDAC based on preoperative CECT images from 56 patients [78]. They developed four models: all texture features, histogram features, run-length features, and co-occurrence features. With all texture features, their model predicted low-grade/high-grade PDAC with 86% accuracy, 78% sensitivity, and 95% specificity. The study enrolled a small number of patients and needs further validation in a prospective multicenter study.

In another retrospective investigation, preoperative clinical radiomics nomograms were examined, employing features extracted from CECT images to differentiate between high-grade and low-grade PDAC and to forecast overall survival (OS) [79]. A model was developed to anticipate histological grade based on radiomics scores extracted from CECT scans (HGrad), yielding an area under the curve (AUC) of 0.75 (95% CI: 0.64, 0.85) in the test cohort and 0.76 (95% CI: 0.60, 0.91) in the validation cohort. However, their model was based on information obtained in part from biopsy, which can be potentially inaccurate given the fact that only a small sample of the tumor is analyzed. We posit that further research on CT images from patients with PDAC confirmed from surgery specimens would yield more precise outcomes.

The most frequent genes that are altered in PDAC are KRAS, CDKN2A, TP53, and SMAD4 [80]. Unfortunately, these markers can only be obtained from biopsy specimens or resected lesions. Developing AI models to facilitate the prediction of these markers in PDAC could provide a better therapy option for the patient, without the need for tumor resection.

Hinzpeter et al. investigated the correlation between CECT-derived radiomics and driver gene mutations in patients with PDAC in a retrospective study that included 47 patients treated surgically for PDAC [81]. They constructed two statistical models to evaluate the predictive ability of CT-derived radiomics features for driver gene mutations. They obtained an acceptable predictive ability for KRAS and TP52 with a Youden Index of 0.56 and 0.67, and mild-to-acceptable predictive ability for SMAD4 and CDKN2A with a Youden Index of 0.5. One limitation of the study was that they did not discriminate between different molecular subtypes of PDAC, which may have important prognostic implications.

In a study conducted by Gao et al., the preoperative prediction of TP53 status was assessed based on MRI radiomics extracted from seven different sequences: dynamic contrast-enhanced (DCE) T1-weighted imaging (pre-contrast, late arterial phase (ap), portal venous phase, delayed phase), T2-weighted imaging (T2WI), diffusion-weighted imaging (DWI), and apparent diffusion coefficient (ADC) [82]. They used the PyRadiomics package to generate 558 two-dimensional (2D) and 994 three-dimensional (3D) images’ features and further constructed models using SVM. The best performance was achieved by the 3D ADC-ap-DWI-T2WI model with 11 selected features, with an accuracy of 91% and an AUC of 0.96. Some limitations of the study were the small number of patients (57) and the fact that the images were acquired on 1.5 T MRI scanners or 3.0 T MRI scanners, and there are limited data on whether there is a clear effect of field strength on radiomics characteristics.

The high expression of Mucin 4 (MUC4), a highly glycosylated membrane-bound protein, promotes tumor cell metastasis and tumor cell resistance to chemotherapy [83,84]. The overexpression of MUC4 is associated with an unfavorable prognosis in patients diagnosed with PDAC [85]. Deng et al. investigated the potential of radiomics derived from magnetic resonance imaging (MRI) to preoperatively predict the status of MUC4 expression in PDAC within a retrospective study involving 52 patients [86]. They extracted two sets of features from the arterial and portal phases, and after univariate analysis followed by minimum redundancy maximum relevance and principal component analysis, features with a cumulative variance of 90% were selected to construct a radiomics model. Additionally, they developed a clinical model. The AUC values for the arterial model, portal model, and combined model were 0.732, 0.709, and 0.861, respectively. The study indicates that integrating the arterial phase model, portal phase model, and clinical model enhances the predictive capacity of MUC4 expression status. Their results reinforce the need to develop models that can analyze multi-phase images. However, the model needs further external validation on larger samples.

Iwatate et al. investigated the relationship between p53 mutations, PD-L1 abnormal expression, and clinicopathological factors in order to construct prediction models [87]. The retrospective study included 107 patients diagnosed with PDAC with CECT images (early and late-phase images) that were used for feature extraction. The AUC for p53 and PD-L1 predictive models were 0.795 and 0.683. The p53-positive and PD-L1-positive groups were associated with poor prognosis (*p* = 0.008, 0.013). Larger prospective studies need to be conducted to support the results of this study.

The prediction of fibroblast activation protein (FAP) expression in patients with PDAC was studied by Meng et al. on a population of 129 patients with pathology-confirmed PDAC [88]. They constructed a multiplayer perceptron (MLP) network classifier based on radiomics features from noncontrast MRI (breath-hold single-shot fast-spin echo T2-weighted sequence and unenhanced and non-contrast T1-weighted fat-suppressed sequences). Their predictive model had an AUC of 0.84 for the training set and 0.77 for the validation set. Their study shows good results using noncontrast MRI images, which could open the door for this modality to become a screening tool in the future. One limitation of the study was that the images were obtained from a single MRI machine, so further validation on images from other machines needs to be conducted.

Knowing the histopathological grade and the expression of different genes before the surgical intervention might change the treatment options for selected patients; however, the histopathological analysis is often determined after surgery. The studies we encountered, which concentrated on genomics and histopathological grading, were conducted on a limited number of cases and require additional validation on larger datasets. However, we anticipate that in the future, AI models could predict the expression of various genes involved in PDAC, leading to personalized therapy becoming a common practice in more healthcare centers.

**Table 6 diagnostics-14-00438-t006:** AI models studied in histopathological and genomic features.

Author	Year	Modality	Scope	Approach	Performance
Cen et al. [79]	2023	CECT	Histopathological grade	Radiomics	AUC 0.76
Deng et al. [86]	2022	MRI	MUC4 expression	Radiomics	AUC 0.861
Hinzpeter et al. [81]	2022	CECT	Correlation with driver gene mutations	Radiomics	Youden Index 0.56 (KRAS), 0.67 (TP53), 0.5 (SMAD4 and CDKN2A)
Gao et al. [82]	2021	MRI	TP53 mutation	Radiomics	AUC 0.96
Meng et al. [88]	2021	MRI	FAP expression	Radiomics	AUC 0.77
Iwatate et al. [87]	2020	CECT	P53 and PD-L1 expression	Radiomics	AUC 0.795 and 0.683
Qiu et al. [78]	2019	CECT	Histopathological grade	Radiomics	Accuracy 86%

#### 4.3.3. Prognosis

In recent years, there has been a notable surge in interest regarding the utilization of radiomic features for predicting the prognosis of PDAC. Selected papers that focused on prognosis can be seen in Table 7. Vezakis et al. evaluated a fully automated pipeline for survival prediction [89]. They trained a nnU-Net (3D-CNN) on an external dataset and generated automated pancreas and tumor segmentations. The radiomics features were extracted from CECT images from a population of 40 PDAC patients. These features were combined with the TNM system staging parameters and the patient’s age. A random forest model was trained to perform an overall prediction over time and random forest classifier for the binary classification of two-year survival. Their results show promise, achieving a mean C-index of 0.731 for survival modeling and a mean accuracy of 0.76 in two-year survival prediction. One limitation of the study was that given the fully automated manner of the model, segmentation inaccuracies may lead to suboptimal radiomics features extraction.

Xu et al. developed an MRI–radiomics nomogram for the prognosis of PDAC that achieved a C-index of 0.780 [90]. The limitations of the study were the small population (78 PDAC patients) and the absence of external validation. The study used images from the arterial phase of T1 sequences with contrast, but further research should also include other sequences.

Qiu et al. developed a prognostic model utilizing a preoperative MRI-based radiomics nomogram for patients diagnosed with PDAC [91]. To mitigate bias stemming from confounding factors, they opted for T2-weighted imaging sequences for the radiomics analysis. They derived a radiomics signature (Rad-score) based on radiomics features that was significantly associated with overall survival (OS) and progression-free survival (PFS). By integrating the Rad-score with clinical parameters, they constructed a clinical–radiomics nomogram. In the development cohort, this nomogram yielded a C-index of 0.814 for OS and 0.767 for PFS. They manually delineated the region of interest (ROI) for radiomics feature extraction, which is time consuming and hard to implement in a clinical setting.

In another study conducted by Li et al. a radiomics–clinical nomogram integrating intra- and peritumoral CT radiomics signature and clinical factors was built in order to assess the recurrence risk of PDAC after radical resection [92]. The radiomics nomogram showed an AUC of 0.764 (95% CI: 0.644–0.859) in the validation test for predicting 1-year recurrence and an AUC of 0.773 (95% CI: 0.654–0.866) in the validation test for predicting 2-year recurrence. We consider that integrating peritumoral CT radiomics features is beneficial for obtaining an accurate prediction of recurrence, because tumoral infiltration in the surrounding tissue could be difficult to assess for the radiologist.

Xie et al. constructed a radiomics nomogram integrating the Rad-score (from CT images) and clinical data that showed better performance of the survival prediction than the clinical model and TNM staging system with a C-index of 0.742 (95% CI: 0.697–0.787) for the training cohort and 0.726 (95% CI: 0.646–0.806) for the validation cohort [93]. The Rad-score was calculated in a manual fashion, which is time consuming and not feasible in a clinical setting.

In an alternative investigation assessing preoperative CT radiomics features for predicting postoperative survival in patients with PDAC, the C-index for survival prediction based solely on clinical parameters was 0.6785. However, the incorporation of CT radiomics features enhanced the C-index to 0.7414, underscoring an improvement in predictive accuracy [94]. This enforces the need to include radiomics features in addition to clinical parameters for postoperative survival prediction in the upcoming studies.

Ni et al. investigated the use of radiomic features extracted from CECT by a pancreas segmentation model to analyze the recurrence of PDAC after radical surgery [58]. The AUC for the nomogram predicting whether the patients will have recurrence after surgery was 0.92 (95% CI: 0.78–0.99), and the C-index was 0.62 (95% CI: 0.48–0.78).

Another study analyzed the use of radiomics features extracted from portal venous CT for predicting surgical portal-superior mesenteric vein (PV-SMV) invasion in patients with PDAC [95]. The radiomics signature had an AUC of 0.848 (95% CI: 0.724–0.971) in the validation cohort. One limitation of the study was the exclusion of patients that received neoadjuvant therapy before surgery. Their model shows promising results and should be tested on larger data sets, including patients who received neoadjuvant therapy, because conventional cross-sectional imaging often cannot precisely identify the extent of the remaining viable tumor [96,97].

The prediction of lymph node (LN) status before surgical resection could have an important impact in the treatment of PDAC. Three papers focused on predicting lymph node involvement [98,99,100]. Bian et al. constructed a prediction nomogram that incorporated the radiomics signature (features extracted from arterial CT scans) and CT-reported LN status to LN metastasis, which had an AUC of 0.75 (95% CI: 0.68–0.82) in the training cohort and 0.81 (95% CI: 0.69–0.94) in the validation cohort [99]. Shi et al. made a radiomics nomogram constructed by radiomics score for T2WI combined with portal venous phase T1TW and MRI-reported LN status that showed an AUC of 0.845 for the training cohort and 0.816 for the validation cohort [98]. Chang et al. constructed a 3D-CNN model that predicted lymph node status with an accuracy of 90% for per-patient analysis and 75% for per-scan analysis [100].

**Table 7 diagnostics-14-00438-t007:** AI models studied in prognosis.

Author	Year	Modality	Scope	Approach	Performance
Vezakis et al. [89]	2023	CECT	Survival prediction	Radiomics	C-index of 0.731
Xu et al. [90]	2023	MRI	Survival prediction	Radiomics	C-index of 0.780
Qiu et al. [91]	2022	MRI	Survival prediction	Radiomics	C-index of 0.814
Li et al. [92]	2022	CECT	Risk of recurrence	Radiomics	AUC 0.764 for 1-year recurrence and AUC 0.773 for 2 year recurrence
Xie et al. [93]	2020	CECT	Survival prediction	Radiomics	C-index of 0.742
Park et al. [94]	2021	CECT	Post-operative survival	Radiomics	C-index 0.7414
Ni et al. [58]	2023	CECT	Recurrenceafter surgery	Radiomics	C-index 0.62
Chen et al. [95]	2020	CECT	Surgical portal-superior mesenteric vein invasion	Radiomics	AUC 0.848
Shi et al. [98]	2022	MRI	Lymph node metastasis	Radiomics	AUC 0.845
Bian et al. [99]	2022	CECT	Lymph node metastasis	Radiomics	AUC 0.81
Chang et al. [100]	2022	CECT	Lymph node metastasis and	3D-CNN	Accuracy 90% for per-patient analysis and 75% for per-scan analysis

In terms of survival predictions, the majority of the papers we reviewed focused on patients with resectable disease. Nonetheless, it is crucial to emphasize that a large number of PDAC patients are diagnosed at an unresectable stage. We advocate for further research to explore multiple treatment options, employing models that predict responses to chemo/radiotherapy.

## 5. Trends and Perspectives

After analyzing multiple papers dedicated to AI models used in radiology detection, segmentation, and classification, we believe that huge progress has been made in developing algorithms that will in the future be implemented in a clinical setting.

The key to increasing the patient’s survivability represents the detection of the tumor at a resectable stage. This can be difficult given the non-specific symptoms of PDAC and the lack of a good delimitation of the small tumors with the normal pancreatic parenchyma on CT/MRI, making it very hard for the radiologist to detect the tumors in the preclinical stage. However, with the new advances in AI models, in the near future, we could implement screening methods for high-risk individuals using low-dose non-contrast CT images or non-contrast MRI images. The advantages of non-contrast acquisitions are the lack of contrast-induced adverse effects and the small acquisition time, which is preferable in the setting of a screening program. AI models could be used as a “second look” in order to enhance the visualization of small and/or isodense tumors, helping the radiologist in detecting PDAC at small dimensions.

Enhancing the automated segmentation of the pancreas and its lesions will benefit researchers by decreasing the time required for tumor segmentation and enabling the inclusion of a greater number of cases in studies. Increasing the number of segmented images will also provide more accurate results in the validation process of the algorithms. The current trend in training models is to include multi-phase images (from at least arterial and venous phases), and with new fast auto-segmentation algorithms, the whole MRI protocol could be included in the process, not just one or two sequences.

PDAC carries such a poor prognosis that in most instances any lesion located in the pancreas that cannot be cleared by the radiologist as benign and is resectable will undergo surgical resection. However, there are a number of benign entities that can mimic PDAC, and with the advances in AI diagnostic algorithms, we believe that in the future, the differential diagnosis of pancreatic lesions will be more accurate so that the patient will not need to undergo any unnecessary surgical procedure.

The trend of analyzing CT and MRI radiomics features and correlating them with the presence of different genes or mutations will improve the management of the patients by deciding whether the patient will benefit from chemo/radiotherapy, especially for borderline cases. Surely, in the future, there will be developed AI models that will predict the histopathological grade, the presence of multiple mutations, and the risk of recurrence based on radiomics features, indicating the best approach for each PDAC case.

However, in order for all these algorithms to become more and more accurate, we posit that multiple institutions need to align with the same objectives and create multidisciplinary and interinstitutional teams so that AI models will be trained and validated on larger data sets, following the same methodology.

## 6. Conclusions

Important advances have been made in the research of artificial intelligence models in PDAC, however multi-institutional collaboration is needed and further validation on bigger datasets should be made in order to construct algorithms that are ready to be implemented in clinical institutions.

## Figures and Tables

**Figure 1 diagnostics-14-00438-f001:**
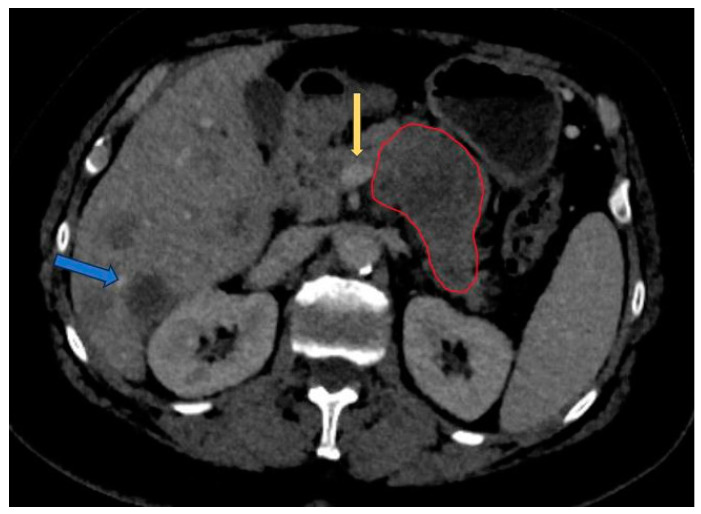
Portal venous phase CT: pancreatic adenocarcinoma involving the corporeal region (red circle) with less than 180° contact with superior mesentery vein (yellow arrow). Multiple liver metastases noted (blue arrow).

**Figure 2 diagnostics-14-00438-f002:**
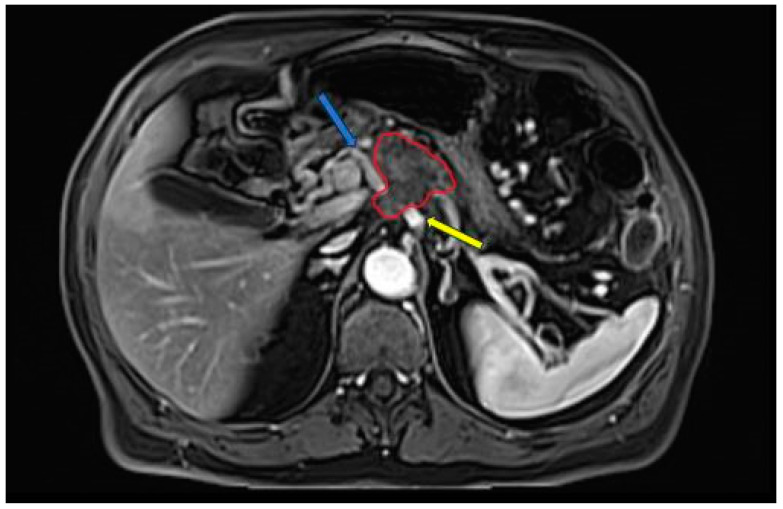
Late-arterial phase 3D T1WI fat-suppressed GRE: pancreatic adenocarcinoma involving the head region (red circle) with less than 180° contact with celiac trunk (yellow arrow) and more than 180° contact with common hepatic artery (blue arrow).

**Figure 3 diagnostics-14-00438-f003:**
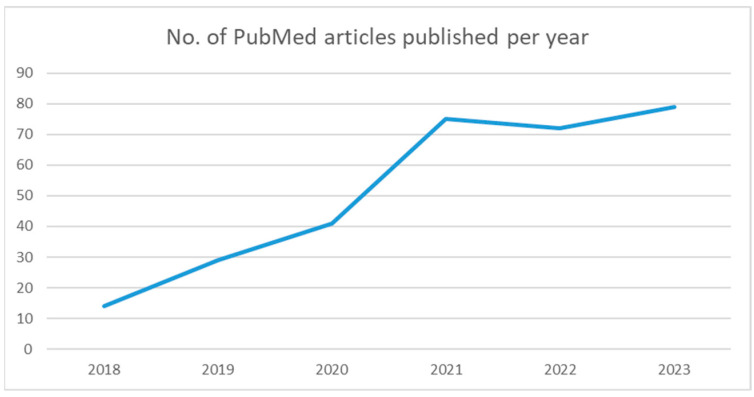
PubMed search for pancreatic adenocarcinoma and artificial intelligence.

**Figure 4 diagnostics-14-00438-f004:**
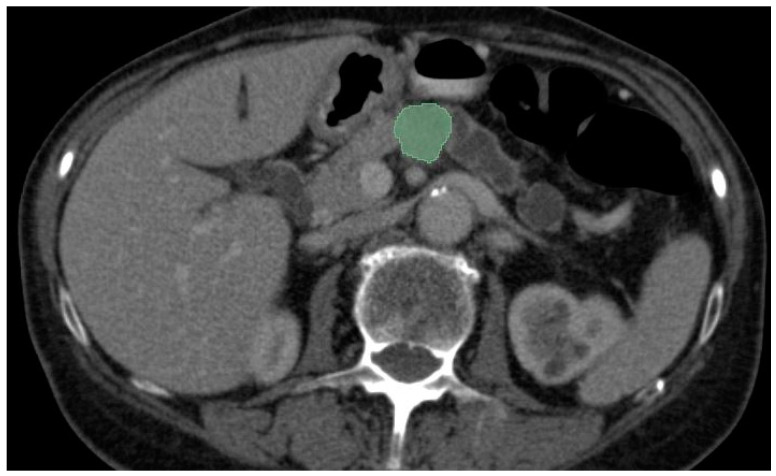
CECT portal venous-phase showing semi-automated corporeal pancreatic tumor segmentation (depicted in green) using “growing-seeds” method.

**Table 1 diagnostics-14-00438-t001:** Free software tools available for AI research.

Name	Website
MaZda ver. 4.6 [18,19,20,21]	https://www.eletel.p.lodz.pl/programy/mazda/ (accessed on 10 February 2024)
3D Slicer ver. 5.6.1 [22]	https://www.slicer.org/ (accessed on 10 February 2024)
PyRadiomics ver. 3.1.0 [23]	https://pyradiomics.readthedocs.io/en/latest/ (accessed on 10 February 2024)
A Computational Environment for Radiological Research (updated on 25 February 2020) [24]	https://cerr.github.io/CERR/ (accessed on 10 February 2024)
LIFEx ver. 7.4.0 [25]	https://www.lifexsoft.org/ (accessed on 10 February 2024)
MONAI ver. 0.8.0 [26]	https://monai.io/ (accessed on 10 February 2024)

**Table 2 diagnostics-14-00438-t002:** Resectability criteria for PDAC depending on location and vascular contacts.

Resectable	Borderline	Locally Advanced
**Arterial** No contact	**Head/uncinate process:**	**Head/uncinate** **process:**
	Tumor contact with common hepatic artery without extension to celiac artery (CA) or hepatic artery bifurcation. Tumor contact with SMA ≤ 180°. Tumor contact with variant arterial anatomy.	>180° SMA or CA
	**Body/tail:** Tumor contact with the CA ≤ 180°	**Body/tail:** >180° SMA or CA or ≤180° CA and aortic involvement
**Venous** ≤180° without contour irregularity	>180° or with contour irregularity/thrombosis resection & reconstruction possible. Tumor contact with IVC	Unreconstructible SMV/PV due to tumor involvement or occlusion (can be due to tumor or bland thrombus)

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
