# Peer review of "Pancreatic Adenocarcinoma: Imaging Modalities and the Role of Artificial Intelligence in Analyzing CT and MRI Images"

_diagnostics, 2024, doi:10.3390/diagnostics14040438_

Round 1

Reviewer 1 Report

Comments and Suggestions for Authors

The manuscript " Pancreatic adenocarcinoma: imaging modalities and the role of artificial intelligence" by Cristian Anghel et al. is a well-written review of challenges in pancreatic ductal adenocarcinoma diagnostics and their possible solutions with artificial intelligence. The review is both informative and valuable, offering significant insights that will greatly benefit physicians and data analysis specialists involved in the field of computer vision.

Major comments

1.     My only major comment concerns the discussion of the results of the reviewed papers. The whole text of section 4 (Discussion) should be integrated into section 3. Currently, Section 3 lacks cohesion and in-depth analysis. While the authors have diligently reviewed significant papers on AI-assisted radiomics, it is crucial to incorporate their perspectives and interpretations of these findings. The Results section needs to better navigate the reader from the overarching themes to specific details. Merely relocating text to subsections is insufficient; the results must be thoroughly presented and elucidated.

2.     My argument, subsequently, continues into the proposal to make a separate chapter on trends and perspectives in which the authors should speculate on the future of AI-assisted PDAC diagnostics.

Minor comments

1.     Alter the title to explicitly indicate its focus on AI analysis of radiology images, as the current title could mistakenly imply coverage of digital pathology.

2.     Revise the sentence starting with “According to the American Cancer Society…” in line 23, considering that the prediction mentioned has already occurred.

3.     The whole paragraph on AI (begins at line 42) and Figure 1 are excessive. Please, shorten the text and present the AI as an effective instrument for data analysis with stating general challenges and trends. It is possibly to the better to exclude the figure from the manuscript.

4.     Rephrase the paragraph on radiomics (begins at line 57). Please, simplify the explanation of how radiological images can predict tumor histology and genetics. Clarify the relevance of detection, segmentation, and classification in clinical contexts.

5.     In line 71, the phrase “applications applicable” should be rephrased to avoid redundancy.

6.     Enhance the graphical representation in Figure 2 using a different graph editing tool. Include a comprehensive title within the graph itself for standalone interpretation.

7.     Avoid explaining the abbreviation CECT twice, as noted in line 80.

8.     Address the issue of duplicate Figure 2 labels. Consider combining all radiology images into a single panel to ensure clear numeration.

9.     In line 121, either introduce the abbreviation US immediately after its full form or avoid using the abbreviation altogether.

10.  Expand the title of Table 1 to provide more context and allow for easier interpretation of its contents.

11.  Begin Subsection 3.1 with a paragraph that explains the challenges of image detection in this context. It is advisable to create a schematic illustration to demonstrate how detection, segmentation, and classification are relevant in clinical settings. Additionally, include a paragraph discussing the challenges of detecting small lesions to aid in understanding the subsequent review content.

12.  Compile a table listing commercially available software for PDAC detection, and consider adding a paragraph to discuss this topic.

13.  Create a table listing publicly available datasets, as this is pertinent to the key points discussed in the manuscript.

14.  In line 225, ensure proper citation formatting for the reference to 3D Slicer.

15.  Correct the grammatical error in line 449, changing “an nnU-Net” to “a nnU-Net”.

16.  Conclude with a summarizing table that encapsulates the findings from papers focusing on the prediction of prognosis.

Reviewer 2 Report

Comments and Suggestions for Authors

The authors perform a review of the highlights in the application of artificial intelligence to clinical aspects of pancreatic cancer. It is well-written, timely, and an interesting read that reviews the most important events over the last 5 years. There are some issues that I think needs to be reviewed prior to publishing, which I support.

Major comment 1: You quickly mention how you performed your search, but it is not detailed enough. Please add how you triaged through the papers to choose which ones you present in more details in the review.

Major comment 2: I am uncertain why you have the imaging section - it is unrelated to AI (other than set the basis for the subsequent sections). I would either add this to the introduction or reposition the imaging modalities elsewhere. It is also a bit illogical to add it here because you use this section to prepare the reader for the other applications. However, you do not prepare for the histology section?

Major comment 3: Please add a large table with all the different classes of papers, or even all the papers you found and classify them into their targeted uses (in the appendix). Then, discuss the generalities of these findings first, then go into detail by saying, "We focused on these papers for XYZ reasons". Otherwise without this information, it just seems like you are highly selecting papers without context.

Overall: Maybe re-sort the paper with:

1. Introduction

2. Diagnostic modalities including radiology + pathology

3. Methods on how you performed the review (and how you selected papers.

4. Figure 2 + large table describing the results of your search.

5. Keep what you wrote for subsequent areas.

Round 2

Reviewer 1 Report

Comments and Suggestions for Authors

The review is excellent! Great work!

Reviewer 2 Report

Comments and Suggestions for Authors

The corrected version is adequate.